# The ABA-LANCL1/2 Hormone-Receptors System Protects H9c2 Cardiomyocytes from Hypoxia-Induced Mitochondrial Injury via an AMPK- and NO-Mediated Mechanism

**DOI:** 10.3390/cells11182888

**Published:** 2022-09-15

**Authors:** Sonia Spinelli, Lucrezia Guida, Tiziana Vigliarolo, Mario Passalacqua, Giulia Begani, Mirko Magnone, Laura Sturla, Andrea Benzi, Pietro Ameri, Edoardo Lazzarini, Claudia Bearzi, Roberto Rizzi, Elena Zocchi

**Affiliations:** 1Department of Experimental Medicine, Section of Biochemistry, University of Genova, Viale Benedetto XV 1, 16132 Genova, Italy; 2Laboratory of Cardiovascular Biology, Department of Internal Medicine, University of Genova, Viale Benedetto XV 6, 16132 Genova, Italy; 3Cardiovascular Theranostics, Istituto Cardiocentro Ticino, Laboratories for Translational Research, Ente Ospedaliero Cantonale, Via Tesserete 48, 6500 Bellinzona, Switzerland; 4Institute of Biomedical Technologies, National Research Council of Italy (ITB-CNR), Via Fratelli Cervi 93, 20054 Milan, Italy; 5Fondazione Istituto Nazionale di Genetica Molecolare, Via F. Sforza 35, 20122 Milan, Italy; 6Department of Medical Surgical Sciences and Biotechnologies, Sapienza University of Rome, C.so della Repubblica 79, 04100 Latina, Italy

**Keywords:** AMPK, PGC-1α, Sirt1, eNOS, NO, ABA, LANCL1/2, mitochondrial proton gradient, cardiomyocyte

## Abstract

Abscisic acid (ABA) regulates plant responses to stress, partly via NO. In mammals, ABA stimulates NO production by innate immune cells and keratinocytes, glucose uptake and mitochondrial respiration by skeletal myocytes and improves blood glucose homeostasis through its receptors LANCL1 and LANCL2. We hypothesized a role for the ABA-LANCL1/2 system in cardiomyocyte protection from hypoxia via NO. The effect of ABA and of the silencing or overexpression of LANCL1 and LANCL2 were investigated in H9c2 rat cardiomyoblasts under normoxia or hypoxia/reoxygenation. In H9c2, hypoxia induced ABA release, and ABA stimulated NO production. ABA increased the survival of H9c2 to hypoxia, and L-NAME, an inhibitor of NO synthase (NOS), abrogated this effect. ABA also increased glucose uptake and NADPH levels and increased phosphorylation of Akt, AMPK and eNOS. Overexpression or silencing of LANCL1/2 significantly increased or decreased, respectively, transcription, expression and phosphorylation of AMPK, Akt and eNOS; transcription of NAMPT, Sirt1 and the arginine transporter. The mitochondrial proton gradient and cell vitality increased in LANCL1/2-overexpressing vs. -silenced cells after hypoxia/reoxygenation, and L-NAME abrogated this difference. These results implicate the ABA-LANCL1/2 hormone-receptor system in NO-mediated cardiomyocyte protection against hypoxia.

## 1. Introduction

Abscisic acid (ABA) is an isoprenoid hormone, conserved during evolution in plants and mammals, with a similar function as a “stress hormone”, regulating the organism’s response to environmental stress [1,2]. In lower Metazoa, ABA is produced by sponge and hydroid cells in response to changes in water temperature and light exposure and regulates functional responses such as water filtration, O_2_ consumption and tissue regeneration [3,4]. In plants, ABA coordinates responses to biotic and abiotic stress, such as changes in light, water, nutrient and oxygen availability [5,6]. In particular, ABA plays an important role in molecular signaling leading to nitric oxide (NO)-mediated stomatal closure in response to several stressors in plants [7]. In mammals, ABA is produced and released by several cell types challenged with different environmental stimuli, including innate immune cells (granulocytes, macrophages, microglia) and keratinocytes, and stimulates cell-specific functions: cell migration, phagocytosis and release of NO [8,9,10]. In particular, the conservation of the key signaling steps ABA/NO in plant cells and human keratinocytes in response to UV-B indicates the significance of this “alarm pathway” as a signaling axis in response to environmental stress, conserved across-kingdom [11].

In mammals, ABA is also involved in glycemia homeostasis: plasma ABA increases after an oral glucose load in healthy subjects [12], and intake of microgram amounts of ABA improves glucose tolerance in rats and in healthy humans undergoing an oral glucose load [13]. The mechanism through which ABA reduces glycemia occurs via an AMPK/PGC-1α/Sirt1-dependent pathway, which activates transcription and expression of GLUT4 and GLUT1 in the skeletal muscle, resulting in a significant increase in insulin-independent glucose uptake in vitro on isolated cells, ex vivo on isolated skeletal muscle and in vivo in rodents and humans [14,15]. ABA also stimulates oxidative glucose metabolism, increases mitochondrial number and respiration and stimulates the expression of uncoupling proteins, both in the skeletal muscle and in brown adipocytes, leading to increased glucose uptake and consumption by these tissues [14].

Two ABA receptors are involved in the metabolic actions of ABA on glucose metabolism: LANCL1 and LANCL2. The mammalian LANCL protein family, which comprises three members, belongs to the lanthionine synthetase family, conserved from bacteria to humans in their sequence but not in their function since mammalian LANCL proteins do not synthesize these modified peptides with antibacterial functions [16]. LANCL1/2 are relatively abundant in all tissues, particularly in the brain, whereas LANCL3 is expressed at very low levels, and its possible function as an ABA receptor has not been explored yet. Conversely, the ABA-binding capacity of LANCL1 and LANCL2, and their role in mediating the above-described functional ABA responses, are well documented. LANCL1 expression in the heart is among the highest in non-neurological tissues and is approximately four times higher compared to LANCL2 expression (Appendix A).

LANCL2 is a peripheral membrane protein located on the intracellular side of the plasma membrane, combining translational features typical of peptide receptors and steroid hormones, i.e., it is coupled to a G-protein, activating adenylate cyclase and is also capable of nuclear translocation [17]. These features may reflect the chemical structure of ABA, which combines an isoprenoid backbone with polar groups, and may also explain the multiple effects elicited by the hormone in different cell types.

At acidic pH values, un-dissociated ABA can cross the lipid bilayer, as occurs in some specialized plant cells [18]. In mammalian cells, transport of dissociated ABA across the plasma membrane is mediated by the anion transporter family AE [19,20].

LANCL1 shares with LANCL2 a sequence identity of approx. 50%, a similar intracellular localization (it is a peripheral membrane protein) and tissue expression pattern [15]. Interestingly, silencing or genetic ablation of LANCL2 in cells or in mice results in the spontaneous overexpression of LANCL1 [15]. Similarly, silencing of LANCL1 causes a significant increase in the expression of LANCL2 in cells [15]. This observation, together with the redundancy of ABA receptors, points to the physiological relevance of the ABA/LANCL hormone/receptor system in mammals. From a functional perspective, LANCL1 binds ABA with a somewhat lower affinity compared with LANCL2, but activates the same signaling pathway (the AMPK/PGC-1α/Sirt1 axis), resulting in similar transcriptional and functional responses (increased glucose uptake and metabolism, mitochondrial respiration and uncoupling) in vitro in rat myoblasts and in vivo in the skeletal muscle of LANCL2^−/−^ mice [15].

Altogether, these observations point to an early (before the separation of Metaphyta and Metazoa) evolutionary origin of ABA as a hormone regulating cell responses to environmental stressors.

The aim of this work was to investigate a possible role of ABA and its receptors, LANCL1 and LANCL2, in the response of cardiomyocytes to hypoxia, in particular, through the production of NO, which has cardioprotective and vasodilating properties [21].

## 2. Materials and Methods

### 2.1. Reagents

All chemicals were purchased from Sigma-Aldrich (Milan, Italy) unless otherwise indicated. 2-cis, 4-trans-Abscisic acid (ABA) was dissolved in water slightly alkalinized with NaOH. 2-deoxy-2-[(7-nitro-2,1,3-benzoxadiazol-4-yl)amino]-D-glucose (2-NBDG) was purchased from Cayman Chemical (Ann Arbor, MI, USA). Antibodies and primers used are listed in Appendix A.

### 2.2. Exposure of H9c2 to Normoxia and Hypoxia

The rat embryonic cardiomyocyte cell line H9c2 cells were purchased from ATCC (LGC Standards s.r.l. Milan, Italy); cells were grown to 70–80% confluence in DMEM (Sigma-Aldrich, Milan, Italy) supplemented with 10% fetal bovine serum (Sigma-Aldrich, Milan, Italy), and 100 U/mL penicillin/streptomycin (Sigma-Aldrich, Milan, Italy). Cells were kept in the incubator at 37 °C in 5% CO_2_ under normoxic conditions. To obtain hypoxic culture conditions, adherent H9c2 cells were washed twice with Hank’s balanced salt solution (HBSS) and then subjected to 30 min of continuous nitrogen flushing. Briefly, 25 cm^2^ culture flasks containing 2 mL of HBSS or 12-well plates containing 0.2 mL of HBSS/well, were placed in a hypoxia incubator chamber (Stemcell Technologies Inc., United States) equipped with a single flow meter set to 10–15 L/minute and flushed with 100% nitrogen for 30 min with a constant flow; then, the hypoxia incubator chamber was kept sealed at 37 °C in the incubator for the time indicated in each experimental setting; at the end of hypoxia, the cells were reoxygenated in the incubator at 37 °C for the times indicated in the legends. Before exposure to hypoxia, H9c2 cells were pre-incubated without or with the NOS inhibitor L-NAME (100 µM) or ABA (100 nM) and kept in the incubator for 30 min.

### 2.3. Measurement of Glucose Uptake and of NADPH Content

A fluorescent glucose analog, 2-deoxy-2-[(7-nitro-2,1,3-benzoxadiazol-4-yl)amino]-D-glucose (2-NBDG, Cayman Chemical, Ann Arbor, MI, USA), was used to measure glucose uptake. H9c2 cells were seeded at 1 × 10^4^/well in 96-well plates; after 6 h, the culture medium was removed from each well and replaced with 100 µL of culture medium without serum. After 15 h, cells were washed in Krebs-Ringer HEPES buffer (KRH) and incubated at 37 °C in 100 µL KRH without additions (control) or with 100 nM ABA (12 wells for each treatment). After 30 min, 100 µL of 50 μM 2-NBDG was added to each well and incubated at 37 °C for 10 min. To stop the uptake, the supernatant was removed, and cells were washed with 200 µL ice-cold KRH buffer. 2-NBDG fluorescence intensity was measured in a microplate reader at an excitation wavelength of 480 nm and an emission wavelength of 540 nm. Non-specific uptake was measured in the presence of 20 µM cytochalasin B and 200 µM phloretin and subtracted from the experimental values [14].

To determine the intracellular NADPH concentration, H9c2 were plated at a density of 1 × 10^6^ cells/well in 6-well plates and cultured in 1 mL of complete DMEM in the presence or absence of 100 nM ABA for 18 h at 37 °C. Then, cells were harvested and lysed in 200 µL of 0.1 M NaOH. Cell extracts were heated at 70 °C for 10 min to hydrolyze NADP^+^ and an aliquot was diluted 20-fold in 10 mM Tris HCl pH 6. Determination of NADPH content was performed as described [22].

### 2.4. O_2_ Consumption

H9c2 cells were cultured in DMEM supplemented with 10% FBS, 2 mM l-glutamine and 100 U/mL penicillin for 48 h without (control) or with 100 nM ABA. Cells were detached with trypsin, washed and resuspended at 3 × 10^5^ cells /mL in 0.1 M KCl, 1 mM EDTA, 2.5 mM EGTA, 5 mM MgCl_2_, 5 mM KH_2_PO_4_, 100 mM TRIS-HCl pH 7.4. O_2_ consumption was measured in a micro-respiratory system (Unisense A/S, Aarhus, Denmark) equipped with an oxygen micro-amperometric electrode; the linear rate of oxygen consumption was measured with an O_2_ electrode for 15 min under stirring at 37 °C in a 2 mL-closed chamber after the addition of 5 mM pyruvate and 5 mM malate.

### 2.5. Nitrite Determination

H9c2 cells were seeded in 6-well plates at 0.5 × 10^6^ cells/well in DMEM supplemented with 10% FBS, 2 mM l-glutamine and 100 U/mL penicillin. After 48 h, the medium was replaced with 1 mL of HBSS containing, or not, 1 mM L-arginine and 100 nM BH4 (as indicated in the legends) and the cells were subjected to normoxia or hypoxia as described above. Nitrites produced from NO and accumulated in the supernatant were detected at 0 and 3 h in 100 μL aliquots with 10 μM 2,3-diaminonaphthalene (DAN) to form the fluorescent product 1-(H)-naphthotriazole as described in [23].

### 2.6. Detection of ABA

H9c2 cells were seeded at 1 × 10^6^ cells in 25 cm^2^ culture flasks in 2 mL of DMEM supplemented with 10% FBS. After 24 h, the medium was replaced with 2 mL of HBSS and the cultures were flushed with nitrogen as described above. After 18 h, cells were detached with a cell scraper and 100 μL-aliquots were recovered for protein content determination; then, 4 volumes of methanol were added to each flask, and the extract was refrigerated at −20 °C. Detection of ABA in the culture extracts was performed by ELISA as described in [12].

### 2.7. Cell Vitality Assays

After hypoxia/reoxygenation experiments in 12-well plates (2 × 10^5^ cells/wells), cell vitality was assessed by Trypan blue exclusion or by MTT reduction assay. The percentage of viable cells was determined after staining with a 0.4% solution of Trypan blue in HBSS by counting at least 100 cells per experimental condition. Before staining, phase-contrast images of the cells were acquired using the Leica Application Suite (LAS, Leica Microsystems, Wetzlar, Germany) software for image analysis, attached to a Leica microscope. Cellular MTT reduction was determined by incubating the cells with 0.5 mg/mL of MTT for 1 h at 37 °C. After washing in PBS buffer, the cells were resuspended in DMSO, and the absorbance at 570 nm was determined in a microplate reader (Fluostar BMG Labtech, Ortenberg, Germany).

### 2.8. Lentiviral Cell Transduction

The lentiviral plasmids pLV[shRNA]-Puro-U6 encoding for a control scramble shRNA (SCR), for the shRNA targeting rat LANCL1 (SHL1) and for the shRNA targeting rat LANCL2 (SHL2) (plasmid ID: VB010000-0005mme, VB181016-1107sen, VB181016-1124zjp), were purchased from Vector Builder (Chicago, IL, USA). Overexpression of hLANCL1 (OVL1) and hLANCL2 (OVL2) was obtained in rat H9c2 cells using pBABE vectors constructed as described in [15], using the empty vector pBABE (Addgene) as negative control (PLV). Lentiviral transductions were performed as described in [15].

### 2.9. qPCR Analysis

H9c2 cells were serum-starved for 12 h, then treated or not with 100 nM ABA for 4 h. Total RNA was extracted from cells using the RNeasy Micro Kit (Qiagen, Milan, Italy), according to the manufacturer’s instructions. The cDNA was synthesized by using iScript cDNA Synthesis Kit (Bio-Rad, Milan, Italy), starting from 1 μg of total RNA, and was used as template for qPCR analysis: reactions were performed in an iQ5 Real-Time PCR detection system (Bio-Rad, Milan, Italy) as described [15]. The rat-specific primers were designed using Beacon Designer 2.0 software (Bio-Rad, Milan, Italy), and their sequences are listed in Appendix A. Statistical analysis of the qPCR was performed using the iQ5 Optical System Software version 1.0 (Bio-Rad Laboratories, Milan, Italy) based on the 2^−ΔΔCt^ method [15]. Values for rat genes were normalized on hypoxanthine-guanine phosphoribosyltransferase-1 mRNA expression. To verify the purity of the products, a melting curve was produced after each run. The dissociation curve for each amplification was analyzed to confirm absence of non-specific PCR products.

### 2.10. Western Blot

H9c2 cells were lysed in a lysis buffer (20 mM Tris HCl [pH 7.5], 150 mM NaCl, 1 mM EDTA, 1% NP40), and a protease inhibitor cocktail was added to the buffer immediately before lysis. The protein concentration was estimated with the Bradford assay, and an equal amount of protein was loaded per lane on a 10% sodium dodecyl sulfate-polyacrylamide gel electrophoresis (SDS-PAGE) gel for protein separation. The specific primary antibodies are listed in Appendix A. After incubation with the proper horseradish peroxidase-conjugated secondary antibody (Appendix A), bands were visualized with Clarity Western ECL Substrate (Bio-Rad, Milan, Italy) and quantified by densitometry using an image analysis system (ImageJ program, Bethesda, MD, USA). The number of phosphorylated proteins was normalized on the respective total protein band.

### 2.11. JC-1 Analysis

H9c2 cells were stained with cationic dye JC-1 (Thermo Fisher Scientific, Waltham, MA, USA), which exhibits potential-dependent accumulation in mitochondria. At low membrane potentials, JC-1 exists as a monomer and produces a green fluorescence (emission at 527 nm), whereas, at high membrane potentials, JC-1 forms aggregates and produces a red fluorescence (emission at 590 nm). Thus, mitochondrial depolarization is indicated by a decrease in the red/green fluorescence intensity ratio [24]. Briefly, H9c2 cells were seeded at 3 × 10^4^ onto µ-slide wells, stained with JC-1 (2.5 µg/mL) for 20 min at 37 °C in a 5% CO_2_ incubator and then imaged live. The red/green ratio was analyzed after background subtraction with the ImageJ software (v1.8.0, National Institutes of Health, Bethesda, MD, USA), using a quantitative analysis based on an intensity measurement of specific selected ROIs.

### 2.12. Statistical Analysis

The results were expressed as mean ± SD. Statistical analysis was performed using GraphPad Prism Software (GraphPad Software Inc., La Jolla, CA, USA). A value of *p* < 0.05 was considered statistically significant.

## 3. Results

### 3.1. Hypoxia Stimulates ABA Release from H9c2 and ABA Stimulates NO Production, Glucose Uptake and O_2_ Consumption

Human granulocytes, monocytes and keratinocytes release ABA when subjected to chemical or physical stressors, and ABA, in turn, stimulates the release of NO [9,10]. Starting from this observation, we investigated whether hypoxia, an important stress factor in the heart, induced the release of endogenous ABA by cardiomyocytes. Rat H9c2 cardiomyocytes were incubated for 18 h under normoxic or hypoxic conditions, and the total ABA content in the cell culture was measured at time zero and at the end of the incubation. Hypoxia induced an approximately three-fold increase in ABA content compared with normoxia (Figure 1A, left panel). This result prompted us to further explore whether ABA, in turn, induced the production of NO in H9c2 cells. NO is an important endogenous mediator of cardioprotection, particularly against hypoxia [21], and hypoxia has been shown to induce the release of NO by rat cardiomyocytes [25]. In line with this observation, H9c2 cells indeed released more NO when exposed to hypoxia compared with normoxia (Figure 1A, central panel, white bars). Incubation with ABA stimulated NO production under normoxia (white bars), with no significant difference between the concentrations tested (10 µM, 1 µM and 100 nM, not shown), and further increased NO generation under hypoxic conditions (Figure 1A, central panel, white bars). Incubation with the NO synthase (NOS) inhibitor L-NAME abrogated the increase of NO in response to hypoxia and also ABA-induced stimulation of NO production in both normoxic and hypoxic conditions, indicating that NOS is responsible for both hypoxia- and ABA-induced NO generation (Figure 1A, central panel, grey bars).

ABA has been shown to stimulate glucose uptake in rodent adipocytes and skeletal myocytes through increased expression and membrane translocation of GLUT4 [12,14,15]. To determine whether ABA also stimulated glucose uptake in cardiomyocytes, H9c2 cells were incubated under normoxia for 30 min in the presence of 100 nM ABA, and uptake of the fluorescent glucose analog 2-NBDG was evaluated. As shown in the right panel of Figure 1A, glucose uptake increased approximately two-fold in ABA-treated H9c2 cells compared with untreated cells. The metabolic fate of glucose in ABA-treated H9c2 cells was its oxidation, both in the Krebs cycle, as inferred from an increase in oxygen consumption, and in the hexose monophosphate shunt, as inferred from an increase in reduced nicotinamide-adenine-dinucleotide phosphate (NADPH) content in ABA-treated compared with control cells (Figure 1A, right panel).

Stimulation by ABA of NO production, glucose transport and O_2_ consumption in H9c2 cells under normoxia prompted the exploration of the signaling pathway responsible for these effects.

### 3.2. Stimulation by ABA of NO Production in H9c2 Cells Occurs Via AMPK and Akt

It has been previously reported that ABA induces phosphorylation of Akt in murine adipocytes [12] and of AMPK in skeletal muscle [14,15]. Both kinases regulate eNOS activity. eNOS is a known substrate of Akt [26], and phosphorylation and activation of eNOS by Akt, inducing an increase in NO production, is indeed observed in myocardial ischemia [27]. AMPK is also involved in the response of cardiomyocytes to hypoxia: its activation improves cell survival, promoting glucose transport and activating eNOS by phosphorylation on Ser1177 via a pathway also involving PGC-1α [28,29,30]. As the AMPK/PGC-1α axis is also a target of the ABA/LANCL system in the skeletal muscle [15], these findings prompted us to explore a possible role for ABA in the activation of Akt, AMPK and eNOS in H9c2 cells. Incubation with 100 nM ABA under normoxia for 2 h resulted in a significant increase in the levels of phospho-Akt (Ser473), phospho-AMPK (Thr172) and phospho-eNOS (Ser1177) compared with untreated cells (Figure 1B, left panel).

To understand the individual role of Akt and AMPK in ABA-stimulated NO release, experiments were performed under normoxia in the presence of specific inhibitors for each kinase (Figure 1B, right panel) and of the AMPK activator metformin.

In the presence of the AMPK-specific inhibitor dorsomorphin (1 µM), ABA-stimulated NO production by H9c2 was abrogated; a similar effect was also observed in the presence of the Akt-specific inhibitor AZD5363 (1 µM). These results demonstrate that eNOS activation by ABA in H9c2 under normoxia occurs via both Akt and AMPK. Metformin (2 mM) instead stimulated NO production, in line with its activation of AMPK.

### 3.3. The ABA-Induced Increase of NO Production Improves Survival of H9c2 under Hypoxia

H9c2 cells were exposed to hypoxia without or with 100 nM ABA, in the absence or in the presence of 100 μM L-NAME. Representative phase contrast images of the cells cultured for 18 h under normoxic or hypoxic conditions are shown in the left panel of Figure 1C. Under hypoxic conditions, a significant reduction of viable cells was observed as compared with control cultures (Figure 1C, right panel, white bars). Incubation with 100 nM ABA significantly increased the percentage of cells surviving hypoxia (Figure 1C, right panel, white bars). Conversely, incubation of the cells with the NOS inhibitor L-NAME abrogated the pro-survival effect of ABA (Figure 1C, right panel, grey bars). This observation, together with the fact that ABA stimulates NO production under hypoxia, indicates a causal role of ABA-induced NO generation in the improved survival of H9c2 exposed to hypoxia.

### 3.4. Overexpression of Either LANCL1 or LANCL2 Activates eNOS Transcription and Function in H9c2 under Both Normoxia and Hypoxia

Recently, several functional responses mediated by LANCL2 were shown to lie downstream of LANCL1, which binds ABA and shares with LANCL2 the ability to activate AMPK transcription and phosphorylation [15]. In rat L6 skeletal myoblasts overexpressing either LANCL1 or LANCL2, a similar activation of the transcription and phosphorylation of AMPK was observed upon incubation with ABA, with increased downstream GLUT4 expression and glucose uptake [15].

In order to understand the individual role of the LANCL proteins in eNOS transcription and function in H9c2, cells overexpressing either LANCL1 or LANCL2 or both, and cells silenced for the expression of both proteins, were generated. Lentiviral infection of H9c2 and the selection of antibiotic-resistant cells resulted in a significant overexpression of LANCL1 (OVL1) or LANCL2 (OVL2) or both proteins (OVL1+2) in the double transfectants by approx. 15, 45 or 35 times, respectively, over control cells (PLV), transfected with the empty vector, as determined by Western blot (Figure 2A).

The overexpression of LANCL1 or LANCL2 increased transcription of AMPK, PGC-1α, Sirt1, eNOS, the arginine transporter CAT-2A and the enzyme GPTCH, involved in the synthesis of BH4, a coenzyme required for NOS activity. Conversely, overexpression of either LANCL1 or LANCL2 reduced transcription of arginase (ARG 2), an enzyme competing with eNOS for its substrate arginine (Figure 2B). Interestingly, transcription of the arginine transporter (CAT-2A) was conversely increased, suggesting that the LANCL1/2-overexpressing cells rely on exogenous arginine for NO synthesis. Overexpression of both LANCL proteins together did not amplify the effect observed in the single transfectants. Treatment of LANCL1/2-overexpressing cells with ABA also did not lead to significant further amplification of the transcriptional effects observed. These results suggest that expression levels of LANCL1 and LANCL2 similarly upregulate the expression of the target genes explored, which include the AMPK/PGC-1α signaling axis and the enzymes involved in NO generation (eNOS, CAT-2A and GTPCH). Interestingly, transcription of arginase, potentially reducing arginine availability to eNOS, was instead reduced. At variance with eNOS, mRNA levels of nNOS were only slightly elevated (approximately two-fold) in LANCL1- or LANCL2-overexpressing cells relative to controls, while iNOS transcription was not modified in LANCL1/2 overexpressing cells.

Transcription of enzymes involved in NAD/P synthesis was also stimulated: NAMPT is the key regulator of NAD^+^ biosynthesis from NMN and ATP, and NADK2 is responsible for NAD^+^ phosphorylation to NADP^+^. Finally, increased transcription of TBC1D1 was observed in LANCL1/2-overexpressing H9c2, similarly to that reported in LANCL1/2-overexpressing L6 skeletal myoblasts [15]. TBC1D1, and its homolog TBC1D4, are key RabGAPs regulating muscle glucose utilization. In murine skeletal muscle, TBC1D1 appears to control exercise endurance [31], and its ablation is associated with high-fat diet-induced diastolic dysfunction, left ventricular fibrosis and cardiac hypertrophy [32]. Increased transcription of TBC1D1 was also observed in L6 skeletal myoblasts overexpressing LANCL1 or LANCL2 [15]. Altogether, these data suggest that LANCL1/2 proteins exert a transcriptional control over TBC1D1 both in skeletal and heart muscle.

Expression and phosphorylation of AMPK and eNOS were also investigated at the protein level by Western blot (Figure 2C). Results obtained confirmed the increase of total AMPK (upper left panel) and eNOS (upper right panel) protein expression and of the respective phosphorylated (activated) forms (lower left and right panels, respectively) in LANCL1- or LANCL2-overexpressing H9c2 cells, with no significant further increase in the double-transfectants, suggesting a similarly positive and non-additive effect of the LANCL1/2 proteins on the activation of AMPK and eNOS.

Finally, the release of NO under conditions of normoxia and hypoxia was investigated in LANCL1/2-overexpressing cells. NO production under hypoxic conditions was approx. 5 and 3 times higher than under normoxia in LANCL1 and LANCL2-overexpressing cells, respectively, i.e., higher than in control PLV cells transfected with the empty vector, where NO production under hypoxia increased two-fold compared with normoxia (Figure 2D). The addition of 100 nM ABA under hypoxic conditions doubled NO release, both in PLV and LANCL1/2-overexpressing cells.

Altogether, these results suggest a similar role of LANCL1 and LANCL2 in the upregulation of the signaling and enzymatic events leading to NO release by H9c2 under hypoxia.

### 3.5. The Combined Silencing of LANCL1 and LANCL2 Reduces eNOS Transcription, Expression and Function in H9c2 under Normoxia and Hypoxia

The fact that overexpression of LANCL1 or of LANCL2 had similar transcriptional and functional effects on the AMPK/Akt/eNOS pathway in H9c2 suggested exploring the effect of their combined silencing on the same target proteins explored in Figure 2.

Silencing of both LANCL proteins was achieved by transfection of cells with a single vector containing sequences encoding shRNAs specific for LANCL1 and LANCL2.

As shown in Figure 3A, knockdown of both LANCL1 and LANCL2 was confirmed by both immunoblot (Figure 3, left and central panels) and qPCR (Figure 3, right panel). Expression of both LANCL proteins was reduced by approx. 90%, and transcription was not increased by ABA treatment, as conversely observed in the control cells, transfected with the vector containing scrambled silencing sequences (SCR) (Figure 3, right panel).

Transcription of the same target genes explored in LANCL1/2-overexpressing H9c2 was investigated in the double silenced cells. Results obtained were almost specular: mRNA levels of AMPK, PGC-1α, Sirt1 and eNOS were significantly reduced compared to control cells and did not increase in ABA-treated cells (Figure 3B), confirming these genes as transcriptional targets of the LANCL1/2 proteins. Conversely, transcription of ARG 2, which was reduced in LANCL1/2-overexpressing cells, was significantly increased in the double-silenced cells compared with control cells, confirming a transcriptional control by LANCL1/2 on these enzymes, competing with eNOS for arginine. Transcription of the arginine transporter CAT-2A, which was upregulated in the overexpressing cells, was instead reduced in the double-silenced cells, as was GPTCH, the key enzyme for BH4 synthesis. Altogether, these data suggest that double-silenced cells are poorly equipped for NO synthesis, with reduced transcription of the arginine transporter, of the key enzyme for the synthesis of the necessary coenzyme BH4, and of eNOS. mRNA levels of nNOS and iNOS were not significantly modified compared with control cells in LANCL1/2 double-silenced cells.

Silencing of LANCL1/2 also significantly reduced transcription of NAMPT, NADK2 and TBC1D1, which were instead increased in overexpressing cells, confirming a transcriptional control exerted by LANCL1/2 on NAD/P-synthesizing enzymes and on the RabGAP TBC1D1.

At the protein level, total and phosphorylated AMPK and eNOS were significantly reduced in the double-silenced cells compared with controls, and protein levels did not increase in ABA-treated cells, as instead observed in control cells (Figure 3C).

Finally, NO release in LANCL1/2-silenced H9c2 was investigated (Figure 3D). Under normoxia, NO production by double-silenced cells, as measured in the presence of excess arginine and BH4, was reduced by approx. 50% compared with control cells, in line with the similar decrease of total and of phosphorylated eNOS observed in these cells. In the absence of added arginine and BH4, NO production by double-silenced cells was an even lower percentage than that measured in control cells (approx. 20%, Figure 3D, left panel), indicating a reduced substrate/coenzyme availability in the double-silenced cells, in line with the reduced expression of the Arg transporter (CAT-2A) and of the BH4-synthesizing enzyme GPTCH and the higher expression of arginase (ARG 2), competing with eNOS for its substrate. Indeed, arginase activity can significantly affect NO synthesis, as indicated by studies on its pharmacological inhibition to boost NO production [33]. Similarly, BH4 availability also profoundly affects NO generation, particularly the “coupling” between arginine consumption and NO vs. ROS production [34]. NO release by LANCL1/2 double-silenced cells did not increase under hypoxia relative to normoxia (Figure 3D, right panel).

### 3.6. Mitochondrial Function Is Conserved in LANCL1/2-Overexpressing, but Not in LANCL1/2-Silenced Cells, Exposed to Hypoxia/Reoxygenation

One of the major factors affecting cell survival to transient hypo/anoxia is the ability of mitochondria to resume normal respiratory function when O_2_ is again available. To investigate the state of mitochondrial function in LANCL1/2-overexpressing or -silenced H9c2 cells after hypoxia, we used the mitochondrial proton gradient (ΔΨ)-sensitive dye JC-1. This fluorescent molecule accumulates within mitochondria and changes its emission from green to red as the ΔΨ increases [24]. As shown in Figure 4A, mitochondrial fluorescence was largely green, with just a trace of red, in LANCL1/2-silenced cells even under normoxia (row A), and the red fluorescence was almost undetectable 3 h after reoxygenation following 30 min of hypoxia (row E).

Conversely, mitochondrial fluorescence was predominantly red in the LANCL1/2-overexpressing cells under normoxia (row C) and remained largely red in cells subjected to the same protocol of hypoxia/reoxygenation (row G). The calculated red/green ratio (Figure 4B) was almost 1 log higher in overexpressing vs. silenced cells, both under normoxia (white bars) and after hypoxia/reoxygenation (black bars). In the presence of the NOS inhibitor L-NAME, the red fluorescence in LANCL1/2-overexpressing cells changed into green, and the red/green ratio was consequently dramatically reduced, indicating a causal role of NO production in the conservation of mitochondrial function in LANCL1/2-overexpressing cells after hypoxia/reoxygenation. H9c2 transformed with the two different control vectors (PLV for the overexpression and SCR for the silencing of the LANCL proteins) had similar red/green ratios under normoxia and after hypoxia/reoxygenation (30 min + 3 h), and the calculated values were in between those of the overexpressing and the double-silenced cells (Figure 4C). Thus, the marked difference in the mitochondrial ΔΨ between overexpressing and silenced cells was not attributable to the different transforming vectors, but to the overexpression vs. silencing of the LANCL1/2 proteins.

Next, the effect of ABA on the mitochondrial ΔΨ was explored in LANCL1/2-overexpressing vs. -silenced cells under conditions of normoxia and of hypoxia/reoxygenation. Two different timings of reoxygenation were explored, 30 min or 3 h, after 30 min hypoxia. Mitochondrial fluorescence (Figure 5A) was largely green in the double-silenced cells, both under normoxia (row A) and after 30 min (row E) or 3 h (row I) reoxygenation following 30 min of hypoxia. The addition of ABA 100 nM only slightly increased the red fluorescence in double-silenced cells under normoxia (row B) and was without evident effect after hypoxia/reoxygenation (rows F and L). In LANCL1/2 overexpressing cells, the red mitochondrial fluorescence under normoxia (row C) was reduced after 30 min reoxygenation (row G) but was again evident after 3 h reoxygenation (row M), indicating the restoration of mitochondrial ΔΨ. In the presence of ABA, the red fluorescence in the overexpressing cells was more visible at the first time point after reoxygenation (row H) and further increased at the second time point (row N). The calculated red/green ratio (Figure 5B) was approx. 1 log higher in the overexpressing vs. the silenced cells under normoxia (white bars) and 3 h after reoxygenation (black bars), and it increased in the ABA-treated cells, particularly in the overexpressing cells, although it did not return to pre-hypoxia values.

As the vectors used to transform cells for overexpression or silencing are different, the red/green fluorescence ratios of cells infected with the empty vector used for overexpression (PLV) and of cells transformed with the vector containing the scrambled sequences used for silencing (SCR) were also compared. Both cell types had similar red/green ratios under normoxia (about 4-fold), and a similarly reduced ratio 3 h after reoxygenation following hypoxia (about 1.5-fold) (Figure 4C), indicating that the vector used for cell transformation could not account for the difference observed in LANCL1/2-overexpressing vs. -silenced cells.

Collectively, these results indicate that in LANCL1/2-overexpressing H9c2 cells, mitochondrial ΔΨ after hypoxia is conserved to a significantly higher degree compared with the silenced cells and further improves after reoxygenation. Moreover, exogenous ABA increases the mitochondrial ΔΨ after reoxygenation in the overexpressing cells, and this effect is reduced in the double-silenced cells.

### 3.7. The Combined LANCL1/2 Silencing Reduces, and Their Overexpression Increases, MTT Reduction, Glucose Uptake and NADP/H Content under Normoxia and Cell Vitality after Hypoxia/Reoxygenation

The data presented above suggest a reduced mitochondrial activity in LANCL1/2 double-silenced H9c2, also under normoxia. We compared the enzymatic reduction of soluble MTT to insoluble MTT-formazan in double-silenced vs. LANCL1/2-overexpressing cells under normoxia. The NADH-dependent reduction of MTT is catalyzed by mitochondrial succinate dehydrogenase; thus, it depends on mitochondrial respiration [35]. Indeed, MTT reduction was almost double in the overexpressing compared with the silenced cells (Figure 6A, left panel).

Glucose uptake under normoxia, as measured with the fluorescent analog 2-NBDG, was also about 40% higher in the LANCL1/2-overexpressing vs. the double-silenced cells, and further significantly increased after pre-incubation with ABA of the overexpressing, but not of the silenced cells (Figure 6A, central panel).

In addition, the NADPH content and the NADPH/NADP ratio were also significantly higher in the overexpressing vs. the silenced cells (Figure 6A, right panel). NADPH is required for NO synthesis and also in the maintenance of the cellular redox balance and antioxidant defense.

Taken together with the results shown in Figure 2, Figure 3, Figure 4 and Figure 5, these data suggest that LANCL1/2-overexpressing cells should be more protected from hypoxia/reoxygenation injury than the double-silenced cells—they produce more NO under hypoxia, have a higher mitochondrial proton gradient, both under normoxia and after hypoxia/reoxygenation and have a higher basal NADPH content and NADPH/NADP ratio.

Thus, we compared cell vitality of LANCL1/2-overexpressing vs. double-silenced cells after hypoxia/reoxygenation. In the microscopic examination, cell vitality was evidently improved in LANCL1/2-overexpressing vs. -silenced cells (Figure 6B, left panel), and trypan blue exclusion confirmed a higher percentage of viable cells in the overexpressing vs. the double-silenced H9c2 (Figure 6B, right panel).

## 4. Discussion

An ABA/NO-mediated signaling pathway likely evolved in aquatic unicellular algae and is believed to have represented a critical step in allowing terrestrial colonization of plants, a key event for subsequent atmospheric oxygen accumulation, dating back some 500 million years [36]. When expressed in the higher plant Arabidopsis, the NOS cloned from the unicellular alga *Ostreococcus tauri* (where it is activated by light exposure) indeed improved the plant response to several environmental stressors [37].

Thus, it comes as no surprise that this signaling axis should be conserved across kingdom, in plants and animals. Indeed, human keratinocytes exposed to UV-B light produce ABA, which activates NO production, a response reminiscent of that of higher plants under excess UV irradiation [11].

Here, we show that the mammalian ABA-LANCL1/2 hormone-receptors system controls fundamental mechanisms in the response of cardiomyocytes to hypoxia/reoxygenation through the activation of the AMPK/PGC-1α axis and of Akt: (i) increased eNOS transcription, expression and phosphorylation; (ii) increased NO generation, also due to the concomitant increased expression of the mitochondrial arginine transporter CAT-2A and of GPTCH, the regulatory enzyme in the synthesis of the coenzyme BH4, necessary for NO generation, and to the concomitant reduction of the expression of arginase ARG 2, competing with NOS for its substrate; (iii) increased glucose uptake and oxidation, leading to higher NADPH content; (iv) conservation of mitochondrial proton gradient (ΔΨ) after hypoxia and increased O_2_ consumption under normoxia. The causal role of NO in the conservation of cell respiration and vitality downstream of the ABA-LANCL1/2 axis is demonstrated by the abrogation of these effects by L-NAME (Figure 7). As L-NAME is a non-specific NOS inhibitor, its effect per se does not allow us to infer which isoform of NOS is mainly responsible for the observed NO-mediated effects on H9c2. eNOS transcription is markedly upregulated (approximately 30-fold) or, conversely, downregulated in LANCL1/2-overexpressing or double-silenced cells, respectively, while mRNA levels of nNOS and iNOS do not appear to be similarly affected (Figure 2 and Figure 3, panels B). This result allows us to tentatively conclude that eNOS may be the principal target of the ABA/LANCL system in H9c2. However, further experiments are needed, e.g., with isoform-specific NOS inhibitors, to draw a firm conclusion, particularly in human cardiomyocytes, which express all three NOS isoforms, each one with specific functional and pathophysiological properties [38].

Downstream of the ABA-LANCL1/2 signaling pathway, both AMPK and Akt appear to be activated at the transcriptional and post-transcriptional levels by phosphorylation (Figure 1, Figure 2 and Figure 3). This apparent redundancy is likely related to the necessity of the heart to integrate different signals and adapt its metabolism to changing conditions of nutrient availability and energy requirements. Thus, Akt, which notably lies downstream of insulin, mediates glucose uptake by the heart under conditions of high blood glucose levels, but AMPK can also stimulate glucose uptake by phosphorylating the same target(s) as Akt (i.e., AS160 and its paralog TBC1D1), in response to an increased workload or to hypoxia [39]. Similarly, both Akt and AMPK can phosphorylate and activate eNOS, leading to increased NO production by cardiomyocytes. NO profoundly affects myocardial function at all levels: electrical transmission, mechano-chemo-transduction (contractility), energy metabolism and myocyte growth and survival [40] and its deficiency is associated with several heart diseases [41]. Indeed, NO replacement therapy is being advocated as a means to counteract endogenous deficiency and improve cardiac performance [42].

Among the target genes downstream of the ABA/LANCL axis, transcription of Sirt1 and NAMPT increased in LANCL1/2-overexpressing and was reduced in double-silenced rat L6 myoblasts compared with control cells [15]. Induction of Sirt1, or overexpression of NAMPT, protects the heart against ischemia/reperfusion injury in rats [43,44]. Conversely, downregulation of Sirt1 is observed in cardiomyocytes from patients with advanced heart failure; indeed, the whole AMPK/Sirt1/NAMPT axis appears to be downregulated in the aging/failing heart [45]. Recently, downregulation of the small interfering RNA miR-217-5p has been shown to protect cardiomyocytes from ischemia/reperfusion injury by restoring mitochondrial function via upregulation of Sirt1 [46]. Thus, the fact that both Sirt1 and NAMPT transcription are upregulated in LANCL1/2-overexpressing cardiomyoblasts is in line with the protective role of the LANCL proteins against hypoxia.

Mitochondria appear to be a target of the ABA-LANCL1/2 system in several cell types. Here we show that H9c2 cardiomyocytes overexpressing LANCL1 and LANCL2 have a higher O_2_ consumption under normoxia and a conserved ΔΨ after hypoxia, as compared with LANCL1/2-silenced cells (Figure 4 and Figure 5). In murine L6 myoblasts, overexpression of LANCL1 or LANCL2 similarly stimulates mitochondrial respiration and the expression of skeletal muscle uncoupling proteins sarcolipin and UCP-3 [15]. In 3T3-derived adipocytes, treatment with ABA stimulates O_2_ consumption and induces transcription of “browning genes”, including UCP-1, and chronic ABA treatment in mice increases mitochondrial DNA content in white adipocytes and UCP-1 expression in brown adipocytes [47]. Indeed, PGC-1α, the master regulator of mitochondrial biogenesis and function, increases at the transcriptional level in LANCL1/2 overexpressing cells (Figure 2B) and is conversely reduced in the double-silenced H9c2 cells (Figure 3B). PGC-1α has been described as a “mediator of the transcriptional outputs triggered by metabolic sensors” [48], and as such, it receives inputs from AMPK and Sirt1, which modify its transcriptional activity via phosphorylation and deacetylation, respectively. Downstream of activated PGC-1α, mitochondrial biogenesis, respiration rate and energy expenditure all increase, particularly in the muscle. PGC-1α also acts by co-activating other transcription factors, including nuclear receptors for several hormones: the thyroid hormone receptor (TR), glucocorticoid receptors (GRs), estrogen receptors (ERs) and estrogen-related receptors (ERRs). In line with its role as a master regulator of energy expenditure, PGC-1α is highly expressed in those tissues that show a high metabolic oxidative capacity, e.g., myocardial and skeletal muscle, brown adipose tissue and brain, and it is induced under conditions that require increased energy production, such as cold, fasting and exercise.

Given its pivotal role in the control of energy expenditure, it is not surprising that a dysfunctional AMPK/PGC-1α/Sirt1 signaling axis should be responsible for reduced muscle energy expenditure, as occurs in aging and in metabolic disorders, such as type 2 diabetes (T2D) [49]. Consequently, pharmacological activation of this pathway holds promise as a new strategy to protect muscle and heart function under conditions that reduce myocyte vitality (aging, hypoxia, diabetes) [28,50,51,52,53].

The results described here could have clinical implications for diabetic cardiomyopathy, a heart condition associated with both type 1 diabetes (T1D) and T2D. The metabolic hallmarks of the diabetic heart are reduced insulin-mediated mitochondrial glucose oxidation and increased free fatty acid uptake, which impairs mitochondrial fatty acid oxidation, resulting in mitochondrial dysfunction, energy depletion and accumulation of toxic lipid metabolites [54]. Defective endogenous ABA has been observed in both T1D and T2D [12,55], suggesting that endogenous ABA deficiency may play a role in the pathogenesis of glucose intolerance due to insulin deficiency or resistance. Indeed, oral ABA has been shown to improve glucose tolerance in borderline and prediabetic subjects [56,57] and to improve insulin sensitivity in murine models of T1D [58]. From the data presented here, we hypothesize that the impaired endogenous ABA function occurring in diabetes could contribute to the reduced resilience of cardiomyocytes to the ultimate stressor of these cells, i.e., hypoxia.

These results warrant further studies aimed at exploring the cardioprotective effect of oral ABA, particularly in diabetic patients. In fact, current strategies to reduce acute ischemia/reperfusion injury advocate the stimulation of glycolysis via AMPK activation, mitochondrial glucose oxidation, NAMPT activity and the elevation of the levels of Sirt1/Sirt3 [43,44], all effects observed downstream of the ABA/LANCL signaling pathway in cardiomyoblasts. In addition, ABA stimulates the release of ATP from human erythrocytes [19], which, in turn, exerts vasodilating effects, which should further improve the recovery of mitochondrial respiration after ischemia.

## Figures and Tables

**Figure 1 cells-11-02888-f001:**
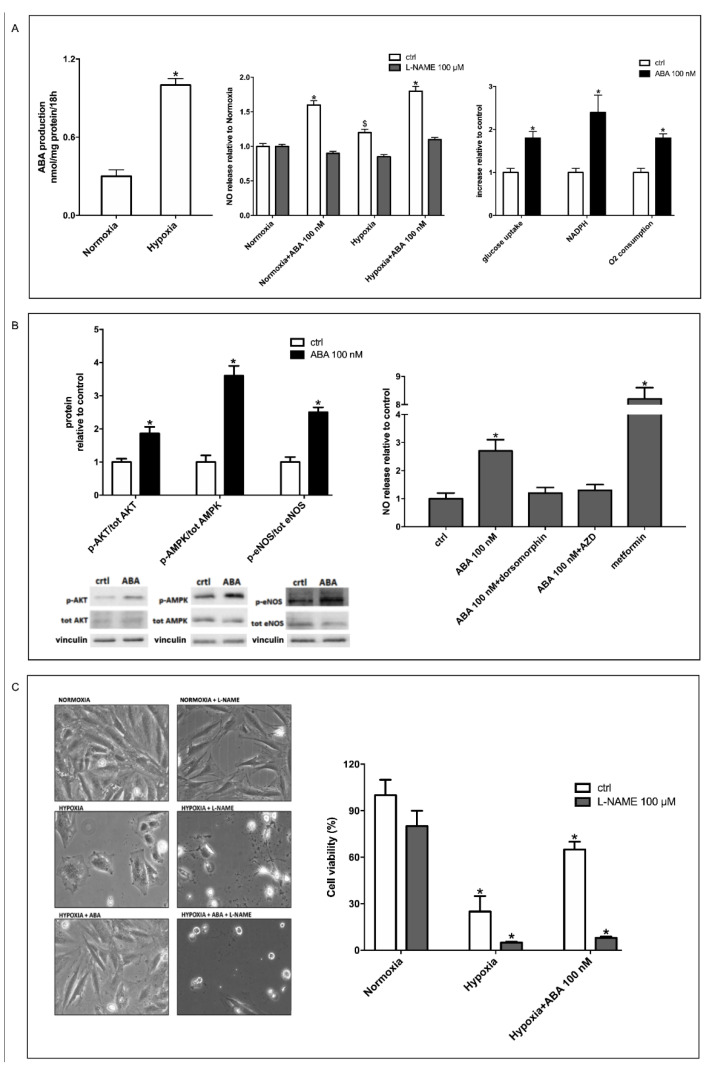
Hypoxia stimulates release of ABA from H9c2 cells, and ABA increases NO production and glucose uptake via AMPK and Akt. H9c2 cardiomyocytes were incubated under hypoxia (30 min under nitrogen flux followed by 3 or 18 h, as indicated in the experiments, in the closed hypoxia incubator chamber at 37 °C), or for the same time under normoxia, without or with 100 nM ABA. The following parameters were explored at the end of the incubation period: (**A**) left panel, ABA content in the culture after 18 h incubation time under normoxia and hypoxia condition; * *p* < 0.01 relative to normoxia by unpaired t-test; central panel, NO released in the culture medium after 3 h incubation, relative to control (cells under normoxia); * *p* < 0.01 and ^$^
*p* < 0.05 relative to control cells under normoxia by unpaired t-test; right panel, glucose uptake, NADPH content and O_2_ consumption in cells incubated without (control, ctrl) or with 100 nM ABA for 18 h, under normoxia. * *p* < 0.01 relative to control cells by unpaired *t*-test. (**B**) Left panel, Western blot analysis of the phosphorylated (p)-to-total (tot) ratio of Akt, AMPK and eNOS in cell lysates obtained after incubation of the cells under normoxia without (control, ctrl) or with 100 nM ABA for 2 h; values are normalized against vinculin, as housekeeping protein; right panel, NO released in the supernatant by cells incubated without (control, ctrl) or with 100 nM ABA for 3 h, in the absence or in the presence of the AMPK inhibitor dorsomorphin (1 µM), the Akt inhibitor AZD5363 (1 µM) or the AMPK activator metformin (2 mM). * *p* < 0.01 relative to control cells by unpaired t-test. (**C**) Left panel, representative phase contrast images of cells cultured for 18 h under normoxia or hypoxia without or with 100 nM ABA and/or 100 µM L-NAME; right panel, cell vitality as determined by Trypan blue exclusion. * *p* < 0.01 relative to normoxia by unpaired t-test. Results shown are the mean ± SD from at least three experiments.

**Figure 2 cells-11-02888-f002:**
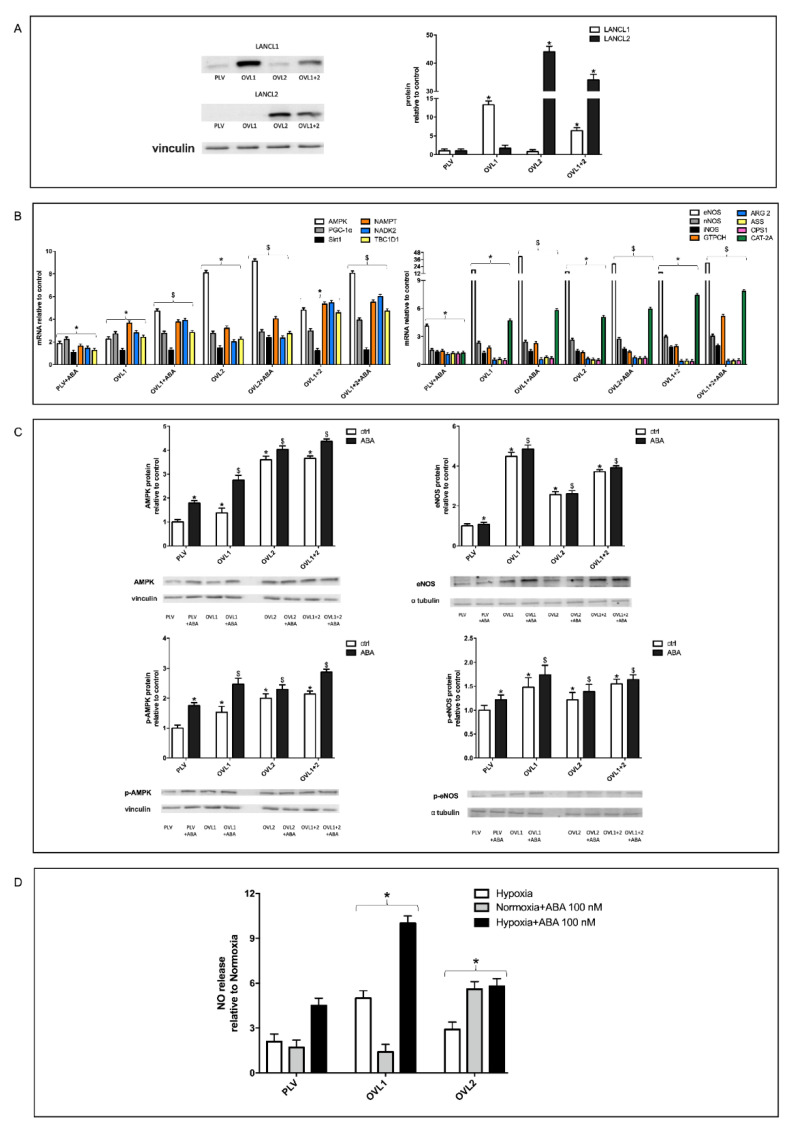
LANCL1 or LANCL2 overexpression stimulates eNOS transcription and phosphorylation and NO production in H9c2 under normoxia and hypoxia. LANCL1 or LANCL2 were individually overexpressed in H9c2 cardiomyocytes by lentiviral infection (**A**), and transcription of putative target genes of the ABA-LANCL1/2 system (**B**), expression and phosphorylation of AMPK and eNOS (**C**) and NO release under normoxia and hypoxia (**D**) were explored. (**A**) Left panel, representative Western blots of LANCL1 and LANCL2 protein expression in cells overexpressing LANCL1 (OVL1), LANCL2 (OVL2), both LANCL proteins (OVL1+2) or infected with the empty vector (PLV); values are normalized against vinculin, as housekeeping protein; right panel, densitometric quantitation of the LANCL proteins expression in the same cell types. (**B**) qPCR analysis of the transcription of the indicated genes in cells overexpressing LANCL1, LANCL2 or both proteins and incubated in the absence or in the presence of 100 nM ABA for 4 h. Results are expressed relative to control cells infected with the empty vector (PLV) and ABA-untreated. (**C**) Western blot analysis of total and phosphorylated AMPK (left panels) and eNOS (right panels) in LANCL1/2-overexpressing cells, treated or not with 100 nM ABA for 1 h. Results are expressed relative to control cells infected with the empty vector (PLV) and ABA-untreated. Values are normalized against vinculin and α tubulin as housekeeping proteins. (**D**) NO release in the supernatant of cells infected with the empty vector (PLV) or overexpressing LANCL1/2 cultured for 3 h without or with 100 nM ABA under normoxia or hypoxia (30 min nitrogen flux, followed by 3 h at 37 °C in the closed hypoxic chamber). Results are expressed as NO release relative to the normoxic condition for each cell type and are the mean ± SD from at least three experiments. * *p* < 0.01 relative to untreated control cells and ^$^
*p* < 0.05 relative to ABA-treated control cells by unpaired t-test.

**Figure 3 cells-11-02888-f003:**
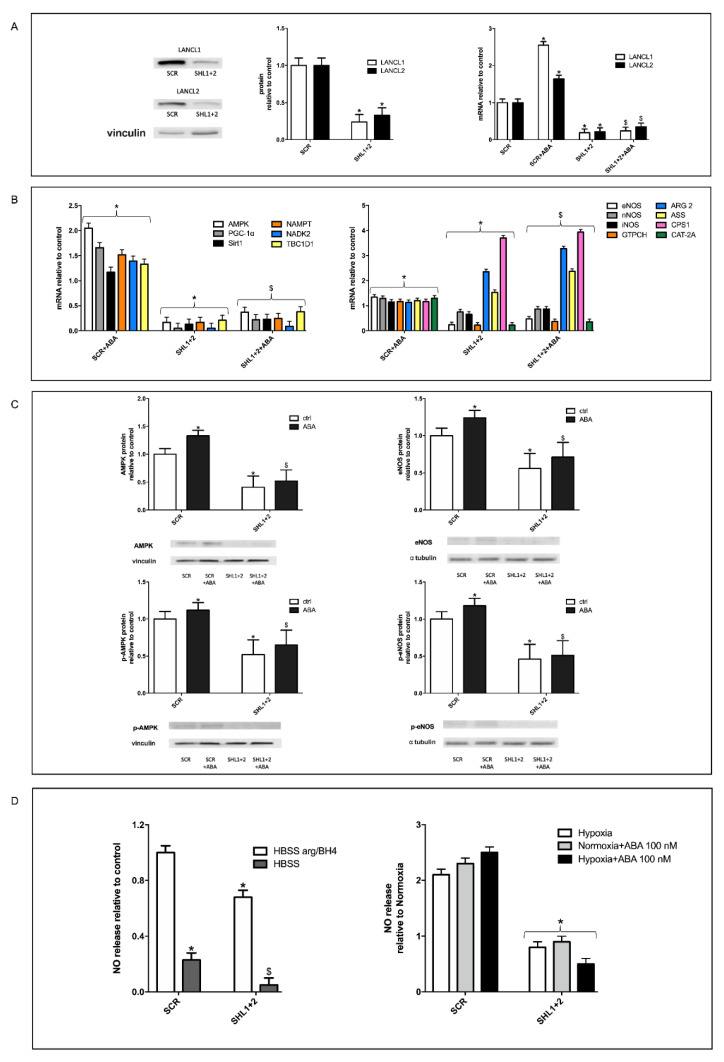
The double silencing of LANCL1 and LANCL2 reduces eNOS transcription, expression and function in H9c2 under normoxia and hypoxia. Stable silencing of both LANCL1 and LANCL2 in H9c2 cardiomyocytes was obtained by lentiviral infection of H9c2 cardiomyocytes with vectors containing the scrambled (SCR) or the LANCL1/2-specific (SHL1+2) silencing sequences (**A**). Transcription of putative target genes of the ABA-LANCL1/2 system (**B**), expression and phosphorylation of AMPK and eNOS (**C**) and NO release under normoxia and hypoxia (**D**) were then explored. (**A**) Left panel, representative Western blots of LANCL1 and LANCL2 protein expression in cells silenced for the expression of both proteins (SHL1+2) or infected with the scrambled sequences (SCR); values are normalized against vinculin, as housekeeping protein; central panel, densitometric quantitation of the LANCL proteins expression in the same cell types; right panel, LANCL1/2 mRNA levels relative to control in LANCL1/2-silenced cells, incubated or not with 100 nM ABA for 4 h. (**B**) qPCR analysis of the transcription of the same genes as in Figure 2B, in cells silenced for both LANCL1/2 proteins and incubated in the absence or in the presence of 100 nM ABA for 4 h. Results are expressed relative to control (SCR), ABA-untreated cells. (**C**) Western blot analysis of total and phosphorylated AMPK (left panels) and eNOS (right panels) in LANCL1/2-double silenced cells, treated or not with 100 nM ABA for 1 h. Results are expressed relative to control (SCR), ABA-untreated cells. Values are normalized against vinculin and α tubulin as housekeeping proteins. (**D**) Left panel, NO release in the supernatant of control (SCR) or LANCL1/2 double-silenced cells cultured for 3 h in HBSS without or with 1 mM arginine (arg) and 0.1 mM tetrahydrobiopterin (BH4). Results are expressed relative to control; right panel, NO release in the supernatant of control (SCR) or LANCL1/2 double-silenced cells cultured under normoxia or hypoxia (30 min under nitrogen flow, followed by 3 h at 37 °C in the closed hypoxic chamber), without or with 100 nM ABA. Results are expressed relative to control cells (SCR) in normoxia without ABA and are the mean ± SD from at least three experiments. * *p* < 0.01 relative to untreated control cells and ^$^
*p* < 0.05 relative to ABA-treated control cells by unpaired t-test.

**Figure 4 cells-11-02888-f004:**
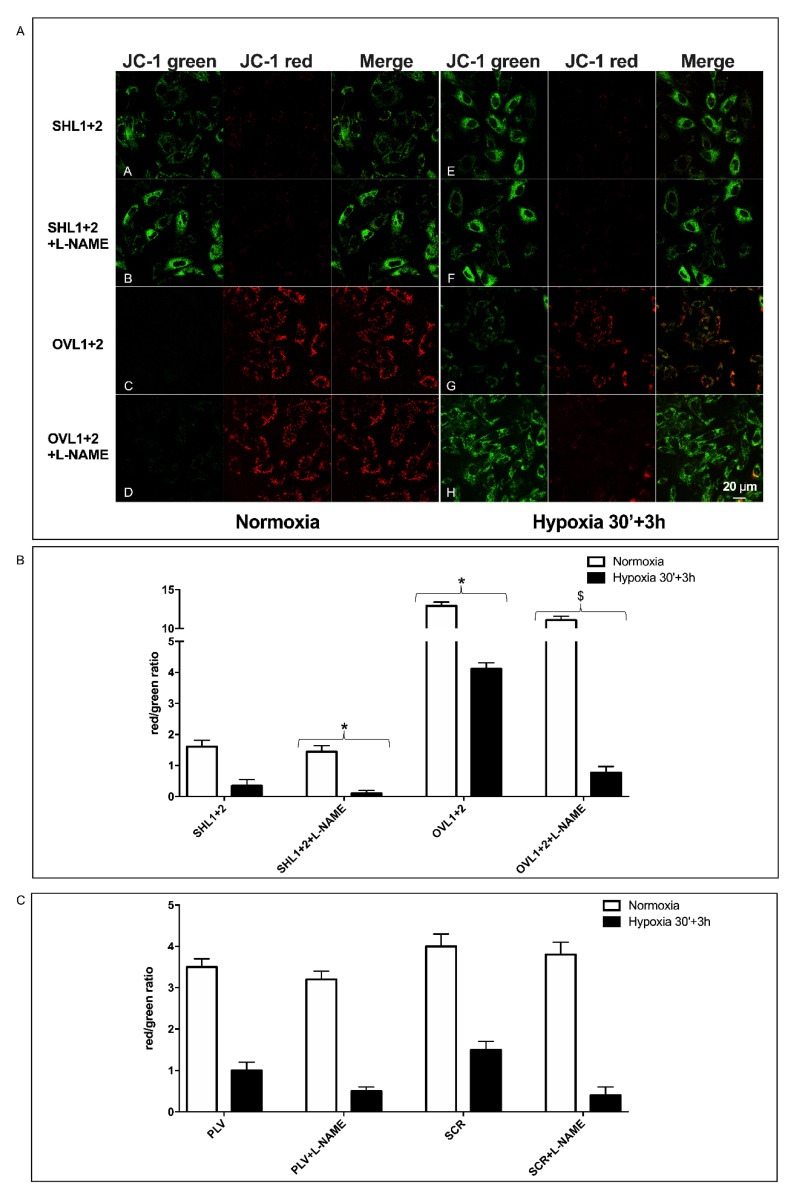
NO-dependent increased mitochondrial proton gradient in LANCL1/2-overexpressing vs. double-silenced H9c2 under normoxia and after hypoxia/reoxygenation. H9c2 cardiomyocytes overexpressing LANCL1 and LANCL2 (OVL1+2), or double-silenced for the expression of both proteins (SHL1+2), were loaded with the mitochondrial proton gradient (ΔΨ)-sensitive fluorescent dye JC-1, which changes its emission from green to red with an increase of ΔΨ (λ_ex_ 488 nm, λ_em_ 527 nm for green fluorescence and at 590 nm for red fluorescence). Cells were cultured without or with 100 µM L-NAME, under normoxia or subjected to 30 min of nitrogen flux followed by 3 h reoxygenation at 37 °C. (**A**) Representative confocal microscopy of the cells. Merged images show JC-1 both as monomer (green) and as J-aggregates (red). Rows A and B, double-silenced cells under normoxia, without (A) or with L-NAME (B); rows E and F, double-silenced cells after hypoxia/reoxygenation, without (E) or with L-NAME (F); rows C and D, LANCL1/2-overexpressing cells under normoxia, without (C) or with L-NAME (D); rows G and H, LANCL1/2-overexpressing cells after hypoxia/reoxygenation, without (G) or with L-NAME (H). (**B**) Red/green fluorescence ratio calculated for the experiments shown in panel A. (**C**) Red/green fluorescence ratio calculated for the controls of each transformed cell type (PLV for the overexpressing cells and SCR for the silenced cells). The mean ± SD of the red/green fluorescence ratio was always calculated in at least 3 microscopic fields. * *p* < 0.01 relative to untreated control cells and ^$^
*p* < 0.05 relative to L-NAME-treated control cells by unpaired t-test.

**Figure 5 cells-11-02888-f005:**
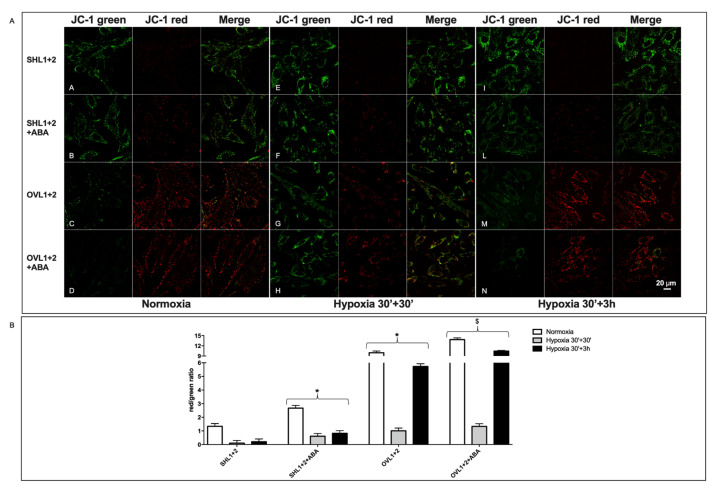
LANCL1/2-overexpressing H9c2 respond to ABA with an increase of the mitochondrial proton gradient after hypoxia/reoxygenation. H9c2 cells overexpressing LANCL1 and LANCL2 (OVL1+2), or double-silenced for the expression of both proteins (SHL1+2), loaded with the ΔΨ-sensitive fluorescent dye JC-1, were cultured without or with 100 nM ABA under normoxia or subjected to 30 min of nitrogen flux followed by 30 min or 3 h reoxygenation at 37 °C. (**A**) Representative confocal microscopy images of the cells exposed to the three different culture conditions, normoxia, hypoxia/reoxygenation (30 min + 30 min) and (30 min + 3 h). Rows A and B, double-silenced cells under normoxia, without (A) or with ABA (B); rows E and F, double-silenced cells after hypoxia/reoxygenation (30 min + 30 min), without (E) or with ABA (F); rows I and L, double-silenced cells after hypoxia/reoxygenation (30 min + 3 h), without (I) or with ABA (L). Rows C and D, LANCL1/2-overexpressing cells under normoxia, without (C) or with ABA (D); rows G and H, LANCL1/2-overexpressing cells after hypoxia/reoxygenation (30 min + 30 min), without (G) or with ABA (H); rows M and N, LANCL1/2-overexpressing cells after hypoxia/reoxygenation (30 min + 3 h), without (M) or with ABA (N). (**B**) Red/green fluorescence ratio calculated for the experiments shown in panel A. The mean ± SD of the red/green fluorescence ratio was always calculated in at least 3 microscopic fields. * *p* < 0.01 relative to untreated control cells and ^$^
*p* < 0.05 relative to ABA-treated control cells by unpaired t-test.

**Figure 6 cells-11-02888-f006:**
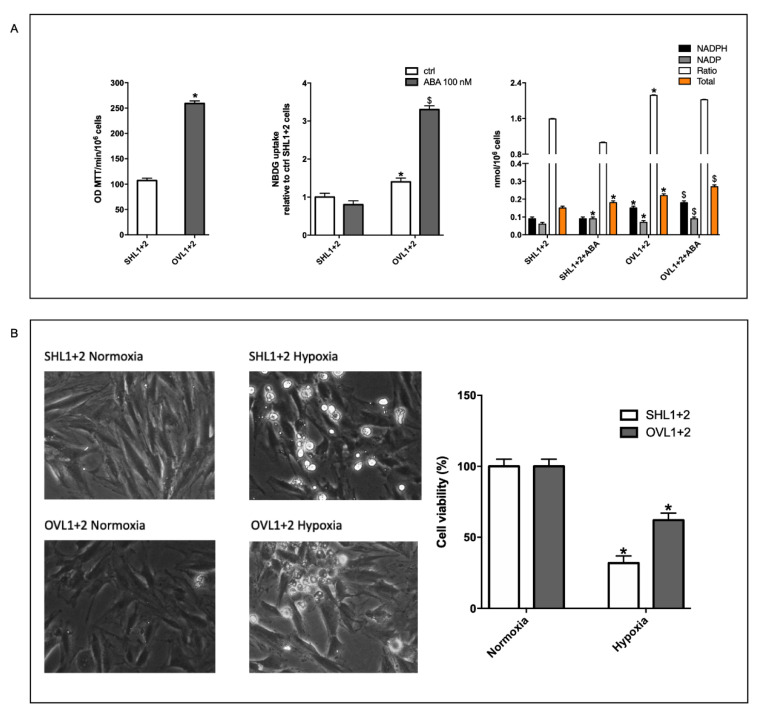
Increased oxidative metabolism and vitality of LANCL1/2-overexpressing vs. -silenced H9c2 under hypoxia. H9c2 cardiomyocytes, overexpressing LANCL1 and LANCL2 (OVL1+2) or double-silenced (SHL1+2) were cultured under normoxia or hypoxia as described in Materials and Methods; oxidative metabolism was explored by the reduction of MTT and the NADPH content (**A**), and cell vitality was evaluated by Trypan blue exclusion (**B**). (**A**) Left panel, after 18 h incubation in complete DMEM in normoxia at 37 °C, cells were washed in PBS buffer and incubated with MTT (0.5 mg/mL) for 1 h at 37 °C. The insoluble formazan crystals were dissolved in 100 µL DMSO, and the absorbance was quantified at 570 nm. Results are expressed as optical density units (OD)/min incubation/10^6^ cells seeded at time zero and are the mean ± SD from at least three experiments. * *p* < 0.01 relative to LANCL1/2-silenced cells by unpaired *t*-test. Central panel, 2-NBDG uptake under normoxia of LANCL1/2-silenced or -overexpressing cells, pretreated or not with 100 nM ABA before the addition of 2-NBDG. Results are expressed relative to the fluorescence of ABA-untreated, double-silenced cells and are the mean ± SD from at least three experiments. * *p* < 0.01 relative to untreated control cells and ^$^
*p* < 0.05 relative to ABA-treated control cells by unpaired t-test. Right panel, NADP and NADPH content in cells incubated in complete DMEM without (control) or with 100 nM ABA for 18 h under normoxia at 37 °C. Results are the mean ± SD from at least three experiments. * *p* < 0.01 relative to untreated control cells and ^$^
*p* < 0.05 relative to ABA-treated control cells by unpaired t-test. (**B**) Left panel, representative phase contrast microscopic images of cells. Right panel, cell vitality as determined by Trypan blue exclusion after 18 h hypoxia and 1 h reoxygenation; at least 100 cells per condition were examined. Results are the mean ± SD of three separate experiments. **p* < 0.01 relative to normoxia by unpaired t-test.

**Figure 7 cells-11-02888-f007:**
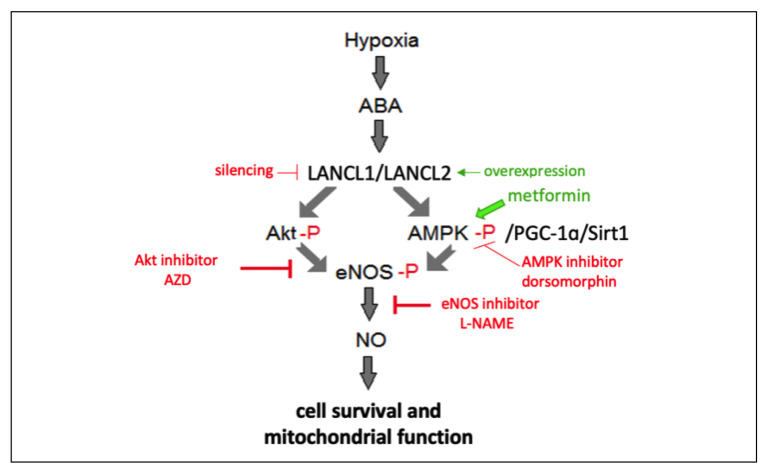
Schematic representation of the role of the ABA-LANCL1/2 system in the response of H9c2 cardiomyocytes to hypoxia. The ABA-LANCL1/2 hormone receptors system is activated by hypoxia in H9c2 cells. Either one of the LANCL proteins can, in turn, activate Akt and the AMPK/PGC-1α/Sirt1 pathway. Both phospho-Akt (Akt-P) and phospho-AMPK (AMPK-P) can phosphorylate and activate eNOS, leading to NO production, which exerts beneficial effects on cell viability and on mitochondrial function (oxidative metabolism and maintenance of the proton gradient). The regulatory pathway was probed by overexpressing or silencing the LANCL proteins, inhibiting or stimulating the activity of Akt and AMPK with specific pharmacologic modulators and by inhibiting NO production via L-NAME.

## Data Availability

Not applicable.

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
