# Peer review of "The ABA-LANCL1/2 Hormone-Receptors System Protects H9c2 Cardiomyocytes from Hypoxia-Induced Mitochondrial Injury via an AMPK- and NO-Mediated Mechanism"

_cells, 2022, doi:10.3390/cells11182888_

Round 1
Reviewer 1 Report
Overall, the findings presented in this manuscript has significant novelty. Although the experimental results came from the cell line H9c2, not the native cells, from the experimental results in this manuscript, the cardiomyocyte protective mechanism by NO via ABA-LANCL1/2 hormone-receptors system is undoubtedly of great significance. So, this work is absolutely groundbreaking for further confirmation on native cells in the future. The experiments were well-designed and done elegantly. The analysis and comparison of the results are rigorous and comprehensive.
Considering to make the above study more comprehensive and rigorous, the following experimental data evidence will be essential, that is, whether nNOS, or even iNOS, plays important role in this putative signaling pathway. In the other words, how much of the NO production observed in the experiment came from eNOS and other isoforms, respectively. Although in the experiments, a non-specific NOS inhibitor, L-NAME, was used to demonstrate the role of eNOS, but apparently insufficient to demonstrate the absence of other isoforms. Of course, if there are research literatures that can confirm that other isoforms are not significantly expressed in the H9c2 cell line, it can also be used as a logically less solid evidence. But the experimental results under the action of relatively specific inhibitors, for example L-NPA for nNOS, are obviously more convincing. It is therefore recommended to supplement this part of the experimental evidence.
In addition, the colorless bar charts are used in the manuscript to display the results of each group. From the reader's point of view, using the colored bar chart is obviously more intuitive.
Also in the third plot in Figure 1A, the bar charts are equally spaced, which causes blurring of the groups.
In figure 2 B, in the colorless bar charts, the spacing of each group is obviously too close. If want black and white, recommend to appropriately increase the spacing of each group. Of course, if it is changed to colored, the intuitiveness of this picture will be greatly improved.
Author Response
Reply to reviewer #1
Considering to make the above study more comprehensive and rigorous, the following experimental data evidence will be essential, that is, whether nNOS, or even iNOS, plays important role in this putative signaling pathway. In the other words, how much of the NO production observed in the experiment came from eNOS and other isoforms, respectively. Although in the experiments, a non-specific NOS inhibitor, L-NAME, was used to demonstrate the role of eNOS, but apparently insufficient to demonstrate the absence of other isoforms. Of course, if there are research literatures that can confirm that other isoforms are not significantly expressed in the H9c2 cell line, it can also be used as a logically less solid evidence. But the experimental results under the action of relatively specific inhibitors, for example L-NPA for nNOS, are obviously more convincing. It is therefore recommended to supplement this part of the experimental evidence.
A specific role for eNOS in the response of H9c2 to stimulation of the LANCL1-2/AMPK/PGC1-α axis is demonstrated at the protein and mRNA level, by the increase of transcription, protein expression and phosphorylation of eNOS.
Research literature indeed indicates that all NOS isoforms are expressed in H9c2 cardiomyoblasts, albeit with different roles. eNOS and nNOS appear to be mainly involved in pro-survival mechanisms activated by hypoxia and/or heat shock, while iNOS activity and expression has been linked to inflammatory responses induced by bacterial stimuli (Zhang, Y.H. F1000Research 2017, 6:742).
In an effort to understand whether transcription of nNOS and/or iNOS are affected by the ABA-LANCL1/2 signaling axis we performed new qPCR analyses on the same mRNA samples used for quantitation of nNOS in LANCL1/2-overexpressing and double–silenced H9c2, treated or not with ABA. Results obtained indicate that nNOS transcription increases approximately 2-fold in LANCL1- or LANCL2-overexpressing cells, relative to PLV-infected controls, and approximately 3-fold in the LANCL1/2 double transfected cells. In LANCL1/2-silenced cells, transcription of nNOS remains unaltered relative to control cells. iNOS transcription, conversely, does not seem to be affected neither by overexpression nor by silencing of LANCL1/2. These results have been added to Figures 2B and Figure 3B, right panels. A comment to these data has been added to the Results (page 8 line 346 and page 10 line 427). The rat-specific primers for nNOS and iNOS have been added to Table S1 of the supplementary data.
However, these results do not allow to rule out a possible role for other NOS isoforms (particularly nNOS) in the ABA/LANCL1-2-induced NO production. For this reason, a comment has been added to the discussion (page 18, line 661), admitting that further investigation is needed to fully appreciate the role, if any, of other nitric oxide synthases in the ABA-mediated effect on cardiomyocytes, particularly in human cells.
In addition, the colorless bar charts are used in the manuscript to display the results of each group. From the reader's point of view, using the colored bar chart is obviously more intuitive.
Colored bars have been added to the graphs in Figures 2B, 3B and 6A (right panel).
Also in the third plot in Figure 1A, the bar charts are equally spaced, which causes blurring of the groups.
Spaces between bar groups in Figure 1 have been increased.
In figure 2 B, in the colorless bar charts, the spacing of each group is obviously too close. If want black and white, recommend to appropriately increase the spacing of each group. Of course, if it is changed to colored, the intuitiveness of this picture will be greatly improved.
Colored bars have been added to Figure 2B.
Reviewer 2 Report
Manuscript ID: cells-1897430
Referee comments
Authors investigated the signal transduction of ABA/LANCL1 and 2/Akt or AMPK/eNOS/NO pathway in cardiomyocyte cell line H9c2 cells subjected to hypoxia/reperfusion and revealed that this pathway is critical for cell survival and sustaining mitochondrial membrane potential and its function.
They performed many well-organized experiments to disclose the key molecules responsible for the pathway step by step. Based on their results, the conclusion is understandable and expected. However, the referee feels several concerns about the results including mitochondrial membrane potential and several parts of the discussion section.
Major comments
In figure 5, compared with figure 4, the red signals of mitochondria were weaker even in OVL1+2 and the levels were almost comparable with SHL1+2 or SHL1+2+ABA. In figure 4 red signals of OVL1+2 in normoxia were very high. The conditions seemed to be comparable, why was it so different? They should present more representative and striking photos in figure 5, although bar graphs of the ratio in figure 5 was almost the same as those in figure 4.
They presented data of mitochondrial potential including normoxia and hypoxia/reperfusion. Furthermore, even in prolonged hypoxia 18 h, OVL1+2 protected cells. According to the ABA/LANCL/NO pathway, ABA or OVL upregulated oxygen consumption due to PGC-1 upregulation and MTT elevation. If so, during prolonged hypoxia, such upregulation of consumption may induce imbalance between oxygen demand and consumption, causing impaired energy metabolism and cell death. However, as their data demonstrated, it could not happen because figure 6 B showed upregulation of cell viability. How do they discuss about this crucial point in the discussion section? Based on the previous studies, they need to discuss and address this issue, i.e., why does the ABA axis protect cardiomyocytes through upregulation of mitochondrial function even in the prolonged hypoxia.
Minor points
Graph presentation depends on authors’ preference. However, a space between two bars in a categorized group seems wider to the referee, for example, the space between white and black bars of pAKT/tot AKT in figure 1A. If the space can be decreased, it will become easy to be seen
Author Response
Reply to reviewer #2
In figure 5, compared with figure 4, the red signals of mitochondria were weaker even in OVL1+2 and the levels were almost comparable with SHL1+2 or SHL1+2+ABA. In figure 4 red signals of OVL1+2 in normoxia were very high. The conditions seemed to be comparable, why was it so different? They should present more representative and striking photos in figure 5, although bar graphs of the ratio in figure 5 was almost the same as those in figure 4.
In fact, upon inspection of the original photographs acquired by the confocal microscope regarding Figure 5, we noticed that a significant loss of color intensity occurred during compression of the file from the confocal dataset to the pdf extension. We are sending attached the original uncompressed images. However, color intensity in the “original” images of Figure 5 (as present in the confocal dataset) was indeed somewhat lower as compared to those of Figure 4. This difference can be attributed to the fact that images were acquired in different experimental sessions, with different laser intensities, which were adjusted in order not to bleach the samples and were thus not always the same in all acquisitions. However, this is exactly the reason why a ratiometric dye, such as JC-1, is better than a single-wavelength one for quantitative measurements, as it allows to calculate a ratio between two fluorescence emissions, which are similarly affected by inter-experimental variations of excitatory light intensity, as well as by other factors such as cell dye retention, photobleaching and variations in cell thickness. Indeed, for data analysis regarding JC-1 fluorescence, the relevant parameter is the ratio between red and green fluorescence. In order to reduce the loss of color intensity occurring after transfer of the originally-acquired images of Figure 5 to other file formats we exported the file with a resolution of 1200 dpi instead of 300 dpi. We believe that in these images the difference in the red signal (Figure 5, OVL1+2 vs SHL1+2), both in terms of intensity and of the number of positive mitochondria, is clearly visible.
In the zip-file of the revised figures we also provide the original confocal images relative to Figures 4 and 5, without compression.
They presented data of mitochondrial potential including normoxia and hypoxia/reperfusion. Furthermore, even in prolonged hypoxia 18 h, OVL1+2 protected cells. According to the ABA/LANCL/NO pathway, ABA or OVL upregulated oxygen consumption due to PGC-1 upregulation and MTT elevation. If so, during prolonged hypoxia, such upregulation of consumption may induce imbalance between oxygen demand and consumption, causing impaired energy metabolism and cell death. However, as their data demonstrated, it could not happen because figure 6 B showed upregulation of cell viability. How do they discuss about this crucial point in the discussion section? Based on the previous studies, they need to discuss and address this issue, i.e., why does the ABA axis protect cardiomyocytes through upregulation of mitochondrial function even in the prolonged hypoxia.
Several factors may contribute to explain this apparent paradox. Firstly, in the absence of oxygen, an increased NO generation stimulates GLUT4 expression and myocyte glucose uptake, alleviating energy deprivation. Allosteric and covalent regulation of key glycolytic enzymes (PFK1, PK) via AMP and AMPK, respectively, likely plays an important role in keeping cells alive until oxygenation resumes. When oxygen becomes available again, respiration and mitochondrial energy metabolism apparently starts sooner in (ABA-treated) LANCL1/2-overexpressing cells and restores the proton gradient more efficiently. Secondly, an increased NO production and low oxygen levels concur to inhibit mitochondrial respiration through the reversible competitive binding of NO to cytochrome c oxidase (Zhang, Y.H. F1000Research 2017, 6:742), preventing mitochondrial damage during hypo/anoxia.
Obviously, these mechanisms are time-sensitive and oxygenation should resume to avoid cell death, the sooner the better; prolonging the time of cardiomyocyte survival to low oxygen supply may allow the success of therapeutic interventions (e.g after AMI).
.
Minor points
Graph presentation depends on authors’ preference. However, a space between two bars in a categorized group seems wider to the referee, for example, the space between white and black bars of pAKT/tot AKT in figure 1A. If the space can be decreased, it will become easy to be seen.
Spaces between bar groups in Figure 1 have been increased. The space between white and black bars has been decreased. Color bars have been added in the most “crowded” panels (Figures 2B, 3B and 6A right panel) to improve clarity.